# Chirality manipulation of ultrafast phase switches in a correlated CDW-Weyl semimetal

Bing Cheng [1] ✉, Di Cheng[1], Tao Jiang [1], Wei Xia[2,3], Boqun Song[1,4], Martin Mootz [1,4], Liang Luo[1], Ilias E. Perakis [5], Yongxin Yao [1], Yanfeng Guo [2,3] & Jigang Wang [1,4] ✉

Light engineering of correlated states in topological materials provides a new avenue of achieving exotic topological phases inaccessible by conventional tuning methods. Here we demonstrate a light control of correlation gaps in a model charge-density-wave (CDW) and polaron insulator $(TaSe_4)_2I$ recently predicted to be an axion insulator. Our ultrafast terahertz photocurrent spectroscopy reveals a two-step, non-thermal melting of polarons and electronic CDW gap via the fluence dependence of a longitudinal circular photo-galvanic current. This helicity-dependent photocurrent reveals continuous ultrafast phase switches from the polaronic state to the CDW (axion) phase, and finally to a hidden Weyl phase as the pump fluence increases. Additional distinctive attributes aligning with the light-induced switches include: the mode-selective coupling of coherent phonons to the polaron and CDW modulation, and the emergence of a non-thermal chiral photocurrent above the pump threshold of CDW-related phonons. The demonstrated ultrafast chirality control of correlated topological states here holds large potentials for realizing axion electrodynamics and advancing quantum-computing applications.

Ultrafast laser excitation has emerged as an important approach to tune and realize dynamical topological phase switch of functionalities, e.g., light-induced Floquet-Weyl phase[1], photo-quench of Weyl semimetal[2] and light-induced Dirac[3] and Weyl nodes[4]. Thus far, the light-control concept has been mainly realized in a nearly free fermion context due to the weak electron-phonon (e-p) and electron-electron (e-e) interactions in related topological materials[5]. An exciting recent research development in condensed matter is the possibility to search for new physics emerging from the interplay between strong correlation and Dirac/Weyl points. For example, strong e-p or e-e coupling in topological materials has the chances to drive a complex phase diagram with symmetry-broken ground states which may embed exotic

topological responses[6,7]. As illustrated in Fig. 1b, a topological Weyl semimetal with quasi-one-dimensional (quasi-1D) lattice could develop multiple correlated states, ranging from a gapped polaron state, an insulating charge-density-wave (CDW) state, to a gapless Weyl semimetal state, depending on the e-p coupling strength and temperature[8–10]. Intriguingly, an insulating CDW state realized in a Weyl semimetal has been argued to be a new avenue to construct a topological axion insulator phase in contrast to previous attempts mainly based on fabricating quantum anomalous Hall sandwich heterostructures[11,12]. This rich phase competition is expected to give rise to more novel physics if these correlated topological phases can be tuned accurately and continuously. Ultrafast excitations of complex

---

[1]Ames National Laboratory, Ames, IA 50011, USA. [2]School of Physical Science and Technology, ShanghaiTech University, Shanghai 201210, China. [3]ShanghaiTech Laboratory for Topological Physics, Shanghai 201210, China. [4]Department of Physics and Astronomy, Iowa State University, Ames, IA 50011, USA. [5]Department of Physics, University of Alabama at Birmingham, Birmingham, AL 35294-1170, USA. ✉e-mail: bcheng2@ameslab.gov; jgwang@ameslab.gov

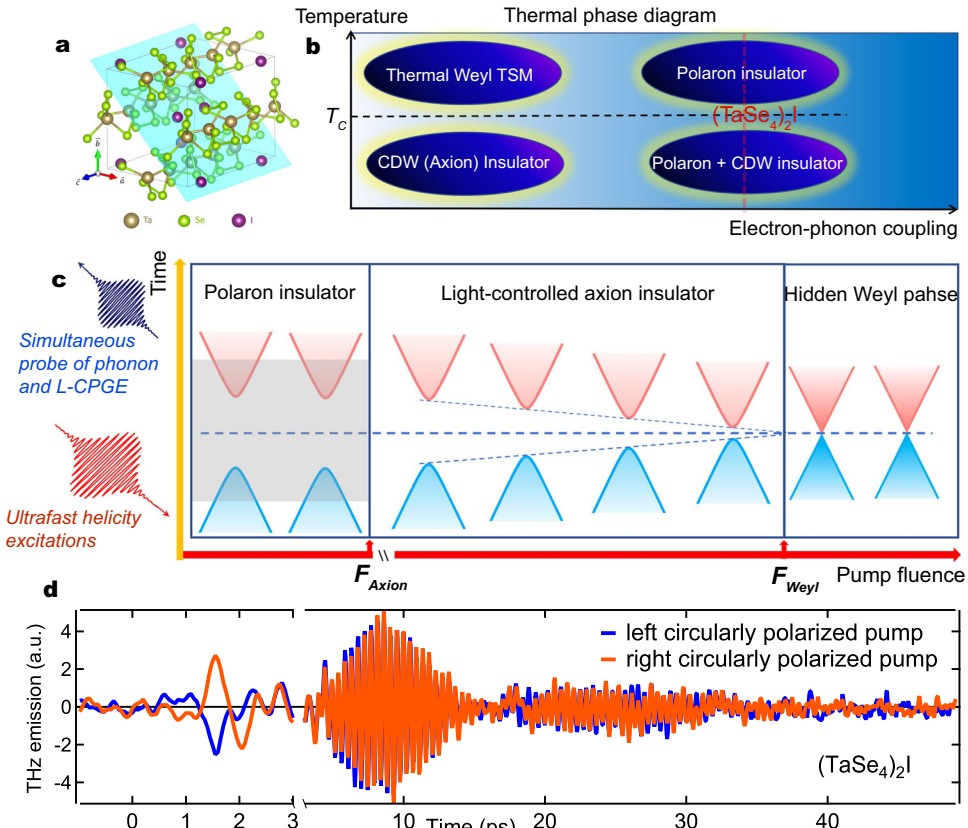

**Fig. 1 | Light-induced phase switches in a correlated CDW-Weyl semimetal.** **a** Lattice structure of $(TaSe_4)_2I$. The highlighted light blue surface is the natural cleavage plane. **b** Phase diagram of topological quasi-1D CDW system as functions of electron-phonon coupling strength and temperature. The strong electron-phonon coupling in $(TaSe_4)_2I$ drives a polaron state which preempts the axion (CDW) insulator phase and the thermal Weyl phase. **c** A schematical illustration of the two-step light melting of polaron state and CDW phase, as well as the light-

induced topological phase switches from an axion insulator state to a hidden Weyl phase. The light shaded area below the polaron gap melting fluence $F_{Axion}$ indicates the possible range of band structure gapped by the polaron gap. $F_{Weyl}$ represents the melting fluence of electronic CDW component. **d** A representative polarized THz emission data along (**k**) after circularly polarized laser excitations with 800 nm central wavelength and 40 fs pulse duration in the single crystal $(TaSe_4)_2I$. The data were taken at 5 K and the pump fluence of $1 \, mJ/cm^2$.

multi-component order parameters and across the correlation gaps provide a fascinating opportunity to generate and distinguish topological phases and tune their competitions. However, to date, experimental demonstration of ultrafast photo-tuned correlated topological states remains scarce.

$(TaSe_4)_2I$ represents a unique setting of correlated topological Weyl semimetals for light-topology quantum control of correlated topological phases. $(TaSe_4)_2I$ is a prototypical quasi-1D Peierls system that develops an incommensurate CDW state below $T_{CDW} \sim 260$ K[13,14]. The formation of electronic CDW gaps wipes out the Weyl node pairs by mixing their chiralities[10,15], which introduces a topological axion term $\theta\mathbf{E}\cdot\mathbf{B}$ into the phase dynamics of the complex CDW order parameter[6]. Here $\theta$ is the effective axion field. **E** and **B** are the electric and magnetic fields. As a result, this unusual gapped Weyl state is predicted to be an axion insulator, a novel topological phase with zero Chern number to realize the long-sought quantized topological magnetoelectric effect and axion electrodynamics[16–18]. Despite the exciting promise, an outstanding challenge to realize such exotic topological phases in $(TaSe_4)_2I$ is the poor screening of the strong $e$-$p$ coupling due to the low lattice dimensionality. Consequently, even in normal state, the charge carriers in $(TaSe_4)_2I$ are always dressed with lattice vibrations and turn into polarons which have vanishing spectral weight and extremely short coherence length[8,9,15,19]. As illustrated in Fig. 1b, the high-temperature phase of $(TaSe_4)_2I$ is actually a polaron insulator, and the low-temperature phase is a polaron plus CDW insulator, preempting both the thermal Weyl phase and the axion insulator

phase[8,9,15,19]. Recent intense debates on if $(TaSe_4)_2I$ is a true axion insulator based on dc magneto-transport measurements may arise from this ubiquitous polaron physics in $(TaSe_4)_2I$[18,20].

In this work, we propose and demonstrate an experimental approach to accurately access, tune, and characterize the genuine exotic topological states in $(TaSe_4)_2I$ hidden by polaron physics. Owing to the ultralow amplitude mode frequency[21,22], the ultrafast responses of electronic and lattice components of the CDW phase in $(TaSe_4)_2I$, i.e., the charge order and the periodic lattice distortion, will have to be extremely decoupled[23,24]. Our experimental approach, polarization-resolved THz emission spectroscopy, is able to simultaneously capture three complementary aspects of the correlated phase tuning: (1) the melting of polaron and CDW gaps is captured by the ultrafast photo-current emission in the sub-ps time scale; (2) the ultrafast response of the periodic lattice distortion is captured by the phonon emission in the tens of ps time scale; (3) and most importantly, $(TaSe_4)_2I$ can be tuned into a topological Weyl semimetal, and its distinctive chiral lattice structure is closely tied to the emergence of chiral currents. It is predicted to intrinsically carry the quantized circular photogalvanic effect (CPGE)[15,25]. The longitudinal component L-CPGE, a parallel or anti-parallel photocurrent along **k** upon different circularly polarized pump, directly correlates with the geometrical chirality that guarantees the existence of topological Weyl fermions (see Methods)[25,26]. Thus, the recovery of geometric band chirality hidden beneath the correlation gaps in $(TaSe_4)_2I$ can be measured directly by the L-CPGE photocurrent as the polaron and CDW gaps are continuously melted.

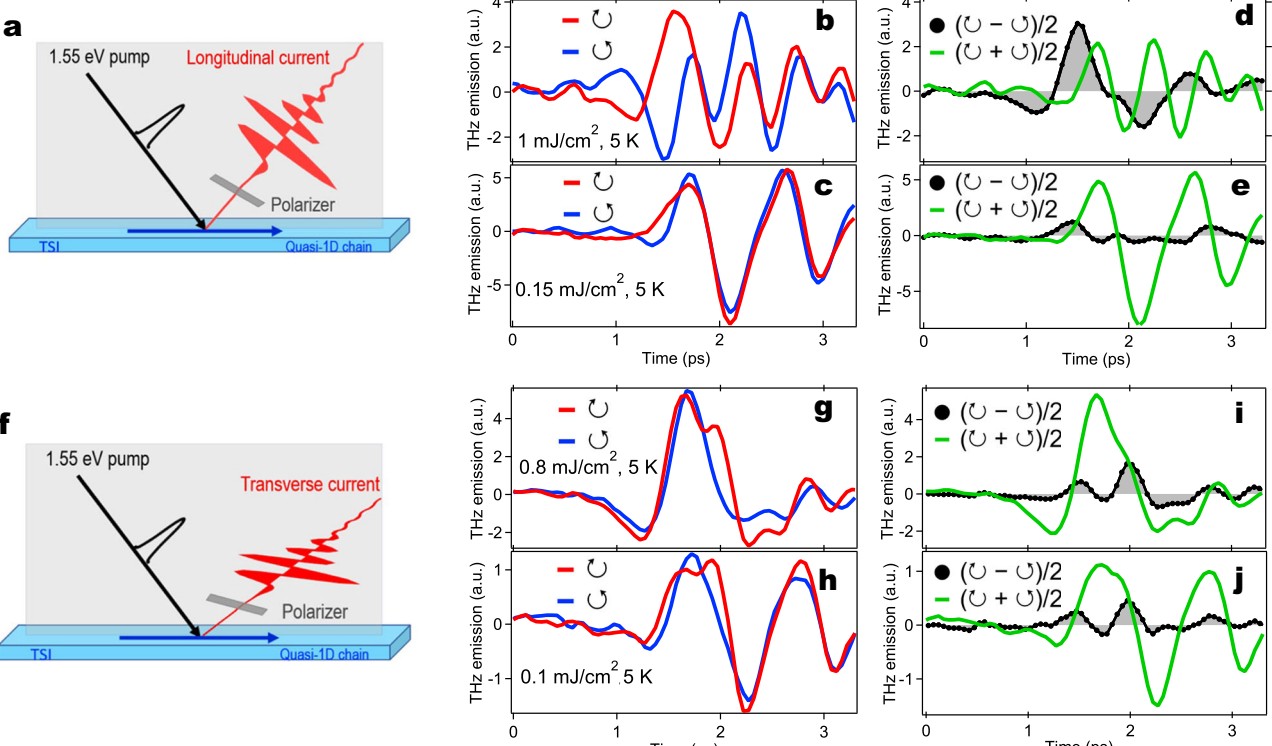

**Fig. 2 | A longitudinal circular photogalvanic current probed by polarized THz emission spectroscopy.** The measurement geometry for the longitudinal (**a**) and transverse (**f**) CPGE photocurrent. The incidence angle of fs laser excitation is 45 degrees. The incidence plane is set parallel to the c-axis chain and perpendicular to the natural cleavage plane. A few THz polarizers are used to selectively probe the desired THz emission components. THz emission in the longitudinal measurement geometry under circularly polarized pump fluence of 1 mJ/cm$^2$ (**b**) and 0.15 mJ/cm$^2$ (**c**). The decomposition of THz emission into the longitudinal CPGE component $(E_\circlearrowright - E_\circlearrowleft)/2$ and the isotropic component of $(E_\circlearrowright + E_\circlearrowleft)/2$ under circularly polarized pump fluence of 1 mJ/cm$^2$ (**d**) and 0.15 mJ/cm$^2$ (**e**). THz emission in the transverse measurement geometry under circularly polarized pump fluence of 0.8 mJ/cm$^2$ (**g**) and 0.1 mJ/cm$^2$ (**h**). The decomposition of THz emission into the transverse CPGE component $(E_\circlearrowright - E_\circlearrowleft)/2$ and the isotropic component of $(E_\circlearrowright + E_\circlearrowleft)/2$ under circularly polarized pump fluence of 0.8 mJ/cm$^2$ (**i**) and 0.1 mJ/cm$^2$ (**j**). The data were taken at 5 K. $\circlearrowright$ and $\circlearrowleft$ represent right and left circular polarization of pump pulse respectively. The light black shaded area is used to highlight the CPGE component.

## Results and discussions

The configurations of our THz emission experiment are illustrated in Fig. 2a, f. The incidence plane is parallel to the chain direction (*c* axis) and perpendicular to the natural cleavage plane (Fig. 1a). These configurations are designed to detect both longitudinal (L-) and transverse (T-) CPGE photocurrents along the directions **k** (Fig. 2a) and **k** × **c** (Fig. 2f), respectively. We show one representative polarized THz emission set of data along **k** in Fig. 1d. This dataset is measured at 5 K and 1 mJ/cm$^2$ under the longitudinal geometry (Fig. 2a). The THz emission temporal profiles exhibit a nearly single cycle oscillation in the first few ps, and the multi-cycle long-lasting oscillations and beatings in the subsequent extended tens of ps time scale. These behaviors clearly disentangle the fast electronic from the slow phononic components via their very different generation and relaxation times. Since the photo-excited hot electrons are expected to quickly relax to Fermi level and band edges within 10s of fs, the much longer-lived, ps photocurrent component should originate from the flow of electrons in the bands near the Fermi level, which intrinsically carries the information of the ultrafast melting of electronic gaps. The even longer-lasting oscillatory component is determined by structural dynamics of the CDW-related zone-folding phonons[21,27]. Remarkably, as the helicity of the optical pump pulse is reversed, the ps photocurrent emission component experiences a π phase shift that indicates a significant L-CPGE photocurrent, consistent with the prediction of CPGE in (TaSe$_4$)$_2$I[15,25]. This behavior represents a benchmark that photoexcitations transfer their net helicities to Weyl fermions flowing in bands with definite chirality[28–30]. In contrast, the phonon emission component is independent of the helicity of the optical pump pulse.

We now present evidence to underpin a light control of ultrafast phase switches via a non-thermal quench of correlation gaps. In Fig. 2b–e, we compare the L-CPGE photocurrent in (TaSe$_4$)$_2$I for two pump fluence 0.15 and 1 mJ/cm$^2$. Here, we emphasize two points. First, we observe a distinct pump fluence dependence that shows the non-thermal effect. Under weak pump fluence (Fig. 2c), the difference of the polarized THz emission profiles between left ($\circlearrowleft$) and right ($\circlearrowright$) circularly polarized pump is small, indicating a weak L-CPGE current. In contrast, under strong pump fluence (Fig. 2b), the difference of THz emission is significantly enhanced. To highlight the L-CPGE photocurrent, we extract the anisotropic, L-CPGE component by calculating the difference between the two helicity contributions $(E_\circlearrowright - E_\circlearrowleft)/2$[31]. The relevant results are shown in Fig. 2d, e together with the isotropic component $(E_\circlearrowright + E_\circlearrowleft)/2$. It is clear that the anisotropic component, L-CPGE, becomes dominant at high pump fluence, while the isotropic current, which probably includes mixed signatures of linear photogalvanic effect (LPGE) current and some initial parts of phonon emission, dominates at low pump fluence. Second, the helicity-dependent photocurrents appear along the wave vector **k** of the pump and become negligible in the traverse direction **k** × **c** for both high (Fig. 2i) and low pump fluence (Fig. 2j). This behavior indicates that the transverse CPGE is not consistent with our observation.

For a more quantitative determination of the fluence dependence, Fig. 3a plots the peak-to-peak magnitude of the L-CPGE photocurrent as a function of pump fluence at 5 and 295 K. Some representative raw traces at 5 K are shown in Fig. 3b. At 295 K (aqua dot, Fig. 3a), we observe an one-step small increase of L-CPGE magnitude which saturates quickly at 0.2 mJ/cm$^2$. In contrast, the magnitude of L-CPGE at 5 K

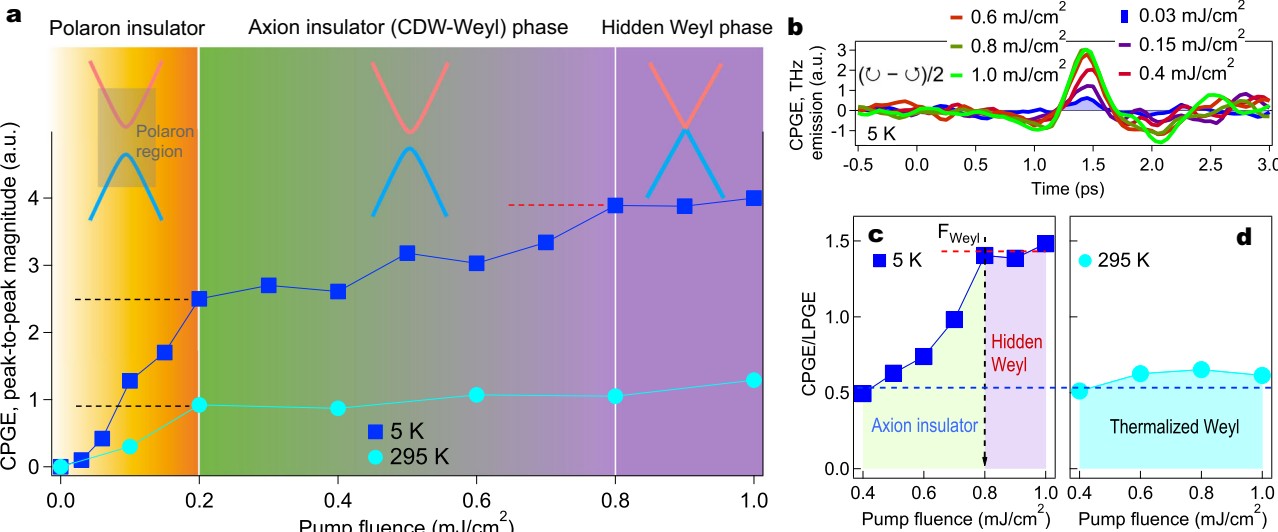

**Fig. 3 | Light-driven phase switch from an axion insulator phase to a hidden Weyl phase. a** The peak-to-peak magnitude of L-CPGE component as a function of pump fluence at 5 K and 295 K. The inset schematically illustrates the evolution of the electronic structure along the chain axis in the light-driven two-step melting process of the polaron state and the axionic CDW phase in $(TaSe_4)_2I$. The gradual evolution of the color along the pump fluence axis imitates the continuous melting of the polaron and electronic CDW gap. The black dash lines label the L-CPGE magnitude at the polaron melting fluence. The red dash line labels the L-CPGE magnitude at the electronic CDW component melting fluence. **b** THz-emission time trace of L-CPGE component $(E_\circlearrowright - E_\circlearrowleft)/2$ at 5 K under pump fluence of 0.03 to 1 mJ/$cm^2$. The 5 K (**c**) and 295 K (**d**) peak-to-peak ratio of CPGE component $(E_\circlearrowright - E_\circlearrowleft)/2$ to LPGE component $(E_\circlearrowright + E_\circlearrowleft)/2$ as a function of pump fluence above 0.4 mJ/$cm^2$. The blue dash line labels the saturated ratio at room temperature above the the polaron melting fluence. The red dash line labels the saturated ratio at 5 K above the electronic CDW component melting fluence $F_{Weyl}$.

(blue square, Fig. 3a) experiences a clear two-step and much larger increase, i.e., after an initial quick increase up to $F_{Axion} \sim 0.2$ mJ/$cm^2$, the L-CPGE emission continues to grow, with a different, relatively slow rate, between 0.2 to 0.8 mJ/$cm^2$. Beyond $F_{Weyl} \sim 0.8$ mJ/$cm^2$, the L-CPGE emission gets saturated. The observed stark different rates or slopes in the build-up of the L-CPGE current between low and high pump fluences features light-induced/tunable phase switches through a light melting of multiple correlation gaps in the ground state of $(TaSe_4)_2I$. The normal state of $(TaSe_4)_2I$ at 295 K is a polaron insulator, and the low-temperature state is a polaron plus CDW insulator[8,9,15,19]. In both low- and high-temperature phases, the polaron gap is found to be ~0.2 eV[9,15]. Therefore, the first melting step at ~0.2 mJ/$cm^2$ below and above $T_{CDW}$ should correspond to the light melting of polarons, which recovers a true axion insulator phase and the thermal Weyl phase respectively. The second melting step at ~$F_{Weyl}$ below $T_{CDW}$, on the other hand, corresponds to the light melting of the electronic CDW order which suppresses the axion insulator state and triggers an ultrafast phase switch into a hidden, gapless Weyl phase. Specifically, the observed L-CPGE photocurrent is shown to be accurately and continuously controlled by the polaron and CDW modulation fluences which give rise to the melting of two (one) correlation gaps below (above) $T_{CDW}$. Therefore, the observation of a two-step pump fluence dependence exclusively at low temperatures indicates the existence of an intermediate photo-tunable state. We interpret such state as the non-thermal and controllable, gapped Weyl or an axion insulator phase prior to a gapless Weyl state.

To highlight the ultrafast phase switch near $F_{Weyl}$, we extract the genuine chiral component of L-CPGE current by measuring the ratio CPGE/LPGE $= (E_\circlearrowright - E_\circlearrowleft)/(E_\circlearrowright + E_\circlearrowleft)$, where $(E_\circlearrowright \pm E_\circlearrowleft)/2$ are the peak-to-peak magnitudes read from the time-domain raw data in Fig. 2d, e. It is worth noting that the ratio CPGE/LPGE provides a direct measurement of the recovery of Weyl band chirality as the melting of correlation gaps, regardless of the pump fluence and temperature used. Fig. 3c and d focus on the pump fluence dependence of the photoinduced longitudinal CPGE/LPGE magnitude above 0.4 mJ/$cm^2$. The room-temperature CPGE/LPGE value is nearly fluence independent, pointing

to a thermal Weyl phase recovered by the one-step light melting of polarons. In contrast, the 5 K magnitude shows a sharp increase above 0.4 mJ/$cm^2$ and a plateau beyond 0.8 mJ/$cm^2$. This distinguishing feature allows us to clearly single out the salient non-thermal phase switch from a true axion insulator phase (hidden by polarons) to a hidden Weyl state. Note that the CPGE/LPGE magnitude contributed from the hidden Weyl state significantly overtakes (three times of) the magnitude from the thermalized Weyl phase, signifying a unique light-driven Weyl state in the low-temperature phase.

Next, we show the selective excitation of the different phonon modes above (below) $F_{Axion}$ that reveals the difference between the light-induced phases under low and high pump fluences. Fig. 4a shows coherent phonon emissions at 5 K for three pump fluences. Interestingly, the long-lasting oscillatory profile shows strong pump fluence dependence. At pump fluence far below $F_{Axion}$, the photocurrent component is extremely weak. The phonon emission dominates the entire THz emission profile. This observation is consistent with the scenario that an insulating polaron state exists close to equilibrium and is robust at low pump fluences. The fast Fourier transform (FFT) spectrum of this coherent signal exhibits a clear resonance at ~1.3 THz (aqua, Fig. 4b), which matches well with the $B_2$ IR phonon mode revealed by the linear optical response of equilibrium state[32,33]. In contrast, at pump fluence far above $F_{Axion}$, a pronounced multi-cycle beating signal becomes significant (see complete dataset in Supplementary Note 5). The FFT spectra of these coherent beatings display multiple dominant peaks centered at ~2.2 THz (black, Fig. 4b). These modes are zone-folding phonons associated with the periodic lattice distortion of the CDW transition in $(TaSe_4)_2I$ (see more dataset in Supplementary Note 6). Our analysis shows they have a $B_3$ IR symmetry which are found negligibly small in the linear response of ground state[34]. At the polaron melting threshold $F_{Axion}$, we see a co-existence of both modes in Fig. 4a, b, while the $B_2$ mode is significantly suppressed comparing to low pump fluences. The suppression of $B_2$ mode at 1.3 THz correlates with the light melting of polaron state, which revives the desired exotic topological phases inaccessible by conventional tuning methods. Figure 4c summarizes the pump fluence

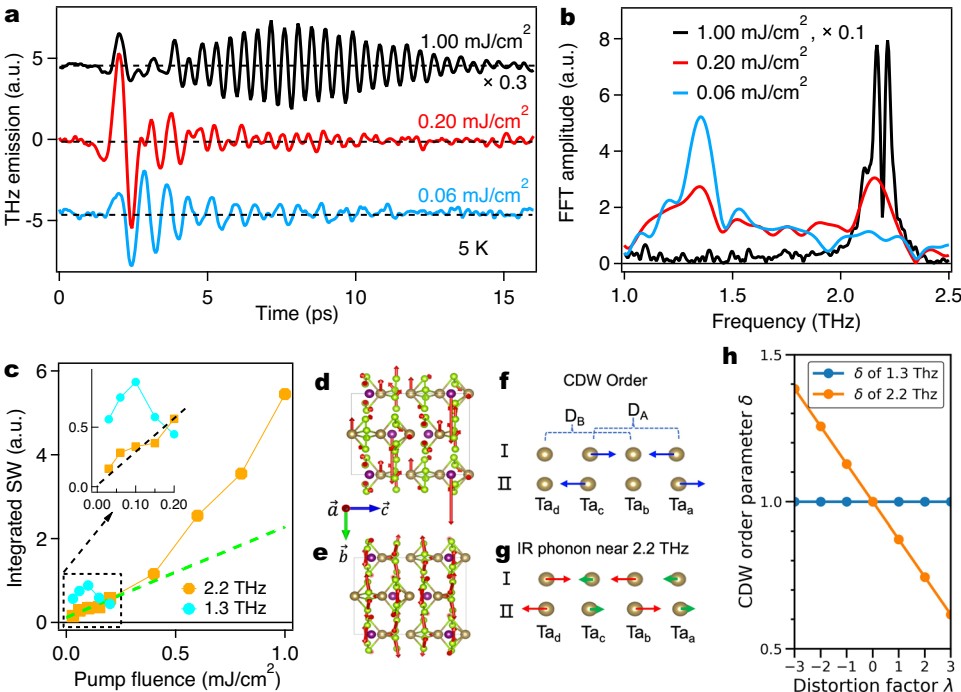

**Fig. 4 | Mode-selective phonon emission as additional evidence for light control of the correlated topological phases in (TaSe₄)₂I. a** THz phonon emission as a function of sampling time delay under pump fluence of 0.06, 0.20 and 1.00 mJ/cm². The linearly polarized excitation laser pulse is set perpendicular to the chain. The THz emission is probed in the geometry shown in Fig. 2a. **b** The magnitude of fast Fourier transformation (FFT) of the phonon emission component. **c** The integrated spectral weight of phonon modes as a function of pump fluence. The inset highlights the fluence dependence of phonon spectral weight at low pump fluence. The crystal structure with eigenvectors of $B_2$ IR phonon near 1.3 THz (**d**) and $B_3$ IR phonon near 2.2 THz (**e**) shown from side (100) view. Atom types of (TaSe₄)₂I are

with the following colors: golden, Ta; green, Se; purple, I. The arrows indicate the direction and magnitude of the atomic displacements of the corresponding phonon mode. **f** Ta atom pair of CDW phase, the blue arrows show the CDW order: atoms displacement from non-CDW phase to CDW phase. $D_A$ is the distance between $Ta_a$ and $Ta_c$ along $c$-axis. $D_B$ is the distance between $Ta_b$ and $Ta_d$ along $c$-axis. **g** Ta atom pair of $B_3$ IR phonon near 2.2 THz. The red arrows show large displacements of $Ta_b$ and $Ta_d$. The green arrows show small displacements of $Ta_a$ and $Ta_c$. **h** Phonon distortion dependence of CDW order parameter $\delta$ for $B_2$ IR phonon near 1.3 THz (blue curve) and $B_3$ IR phonon near 2.2 THz (orange curve), respectively.

dependence of the coherently excited phonon spectral weight at $B_2$ and $B_3$ symmetries. Most intriguingly, the $B_3$ modes exhibit a clear linear to nonlinear transition (Fig. 4c) near $F_{Axion}$ that strongly correlates with the suppression of $B_2$ phonon mode. This crossover indicates the light modulation of the CDW order, which is equivalent to the light engineering of the axion insulator phase in a Weyl-CDW system, starts to dominate the ultrafast response as the polaron state is melted. Furthermore, the switch between the polaron- to CDW-related coherent phonons in Fig. 4c strongly correlates with the emergence of a significant non-thermal chiral current as CPGE/LPGE amplitude in Fig. 3c. Therefore, the selective excitation of the $B_3$ ($B_2$) mode above (below) $F_{Axion}$ validates our conclusion that the light-induced topological phases like the axion insulator state and the hidden Weyl phase are distinguished from the equilibrium and/or thermalized Weyl phase by different collective oscillations associated with it. It is worth noting that above the melting fluence of the electronic charge order, the spectral weight of $B_3$ modes still increases. This inconsistency of melting fluence corroborates with their decoupling nature after photoexcitation on the time scale of electron-phonon thermalization. In some extreme circumstances such as in the prototype quasi-1D CDW system $K_{0.3}MoO_3$, the melting fluence of the periodic lattice distortion is ten times higher than the melting fluence of the electronic density modulation[35].

To put the mode-selective coupling in the CDW and non-CDW phases on firm ground, we carry out IR phonon spectra calculations based on density functional theory (DFT)[36] (see Methods and Supplementary Note 8). In Fig. 4d, e, we illustrate the eigenmodes of the $B_2$ IR phonon near 1.3 THz and the $B_3$ IR phonon near 2.2 THz in the CDW phase. To facilitate the study of phonon coupling to CDW

order, we define a CDW order parameter as $\delta = (D_A - D_B)/(D_A^0 - D_B^0)$, where $D_A$ and $D_B$ are Ta atom pair distances as schematically illustrated in Fig. 4f, g, and $D_A^0$ and $D_B^0$ denote the equilibrium distances[10]. As shown in Fig. 4h, the CDW order parameter $\delta$ can be modulated linearly with the $B_3$ phonon displacement, while it is inert to the $B_2$ mode. In consistency with the experiments, this selective coupling suggests that the strong $B_3$ mode oscillations arise from the light disturbance of the CDW phase. We further estimate the sensitivity of the $B_2$ IR phonon to different phases by comparing the intensity in the CDW and non-CDW phases. The calculated $B_2$ (nearly two-fold degenerate) phonon intensity in the CDW phase is found to be close to that of the associated phonon in non-CDW phase (see Supplementary Note 9), in consistency with the observation that the $B_2$ mode is decoupled from the CDW order. The insights of the mode-selective phononic coupling in strongly correlated materials expands upon recent findings related to light-induced phononic symmetry changes observed in various materials, such as topological insulators[37–40], semimetals[3,4], superconductors[41] and photovoltaic semiconductors[42,43].

In summary, we have discovered a light-induced phase switch in a correlated topological material and demonstrated an ultrafast chirality control enabled by fs circularly polarized excitations. The observation of the longitudinal CPGE photocurrent in (TaSe₄)₂I reveals the presence of geometrical chirality associated with the multiple light-driven phases, at intermediate and high pump fluence, respectively. Our results open opportunities to discover and control hidden topological quantum phases that are inaccessible via "slow" thermodynamic tuning methods. The revealed dynamic strategy of ultrafast topological phase switches also warrants an in-depth investigation to elucidate

the dynamic signatures of axion quasiparticles. Our work provide a universal control principle to induce quantum geometry and non-trivial topology of electronic bands in strong correlated materials, which connects the entire field of strongly correlated electronic materials with topological phenomena.

## Methods

### THz emission spectroscopy

The setup of THz emission spectroscopy is driven by a Ti-sapphire amplifier with 800 nm central wavelength, 1 kHz repetition rate and 40 fs pulse duration. Technical details can be found elsewhere[44–47]. The laser is split into two beams. One is used to photoexcite the $(TaSe_4)_2I$ sample at a 45 degree incidence angle. The incidence plane is set parallel to the $c$-axis chain and perpendicular to the natural cleavage plane (see Figs. 1b, 2a). The other beam is used to measure the emitted THz pulse in the time domain by electro-optic sampling through a 1 mm ZnTe crystal. A quarter waveplate is placed in the pump beam in order to change the polarization, that is, helicity, of the pump beam. In front of the THz electro-optic sampling crystal ZnTe, a few THz polarizers are used to measure the desired polarization component of THz photocurrent emission, either parallel or perpendicular to the plane of incidence (highlighted in Fig. 2a, f).

### THz detection bandwidth

We use a laser system with 800 nm central wavelength, 1 kHz repetition rate and 40 fs pulse duration to generate and detect THz pulse. We had tested the detection bandwidth of our system. We used a ZnTe crystal to generate THz pulse, and then let the THz pulse go through a 1 mm pinhole. Finally the THz pulse is detected by a second ZnTe crystal. We found our THz detection bandwidth is optimized between 0.5–2.5 THz. We provide calibration data in Supplementary Note 7. Therefore, it is not surprising to see that our THz system does not resolve any 0.1 and 0.2 THz amplitude modes since they are outside of our detection bandwidth[21,22,48].

### Sample growth

The $(TaSe_4)_2I$ crystals were grown by chemical vapor transport method. Starting materials of Ta powder (99.9%, Macklin), Se granules (99.999%, Aladdin) and Iodine (99.99%, Aladdin) were mixed in a molar ratio of 1:4:1, loaded into an alumina crucible, and then was sealed in a quartz ampoule under partial argon atmosphere. The assembly was heated up in a two-zone furnace with high and low temperature sides being 550 °C and 400 °C, respectively. The sample was slowly heated up to the target temperatures in 5 h, and stayed for 200 h before naturally cooling down. Finally, the assembly was taken out from the furnace. Strip like crystals could be found in the cold zone with a typical size of ~$1 \times 1 \times 5$ mm$^3$.

### L-CPGE in $(TaSe_4)_2I$

$(TaSe_4)_2I$ has a quasi-1D body-centered tetragonal chiral lattice structure in SG97 (I422) that lacks improper rotation symmetries, such as inversion or mirror symmetry[15]. Band structure calculation shows there are 24 pairs of Weyl points near the Fermi energy. All these Weyl points are located within a close vicinity of the $k_z = \pm\pi/c$ planes[15]. The CDW phase transition will gap all Weyl points below the Fermi energy. Because the Weyl points with opposite chiral charges appear at different energies, $(TaSe_4)_2I$ is predicted to exhibit circular photogalvanic effect (CPGE). Most remarkably, the CPGE in $(TaSe_4)_2I$ could be quantized, which is the bulk signature of topological chirality[15,25]. The longitudinal component L-CPGE, a parallel or anti-parallel photocurrent along **k** upon different circularly polarized pump, directly correlates with geometrical band chirality which guarantees the existence of topological Weyl fermions[25,26].

### The equilibrium phase diagram of $(TaSe_4)_2I$

The insulating polaronic state at equilibrium is well-established by several static and time-resolved ARPES measurements[8,9,15,19]. The sample growth process of the $(TaSe_4)_2I$ crystal will more or less introduce some defects and self-doping effects which drive the Fermi level a little further away from the CDW gap region. As shown in Fig. S15 of the referred Nature Physics paper[15], above and far below the CDW transition temperature $T_{CDW}$, one can always observe a robust energy gap near the Fermi energy. This gap is referred to the polaron gap and is induced by strong electron-phonon coupling. The real CDW gap (~0.2 eV) is actually below the polaron gap. A more exciting and persuasive introduction of polaron physics in $(TaSe_4)_2I$ is given by a recent time-resolved ARPES work[19]. Note that the size of the polaron gap is basically the same both at room temperature and far below $T_{CDW}$. This feature could exclude some other possibilities, such as a fluctuating CDW above $T_{CDW}$. It is hard to believe that in the deep and well-developed CDW phase (like at 5 K), a fluctuating CDW still exists to generate a pseudogap just above the well-developed CDW gap. Therefore, our phase diagram of $(TaSe_4)_2I$ at equilibrium precisely describes the physics in this model quasi-1D CDW system.

### Other THz generation mechanisms in $(TaSe_4)_2I$

Besides the longitudinal circular photogalvanic effect, other mechanisms, such as the photo-Dember effect, photon drag effect, and photo-thermoelectric effect, could be able to contribute THz photocurrent emission too. With further measurements, we could exclude all these sources and underpin the main contribution of the longitudinal circular photogalvanic effect. THz emission may arise from the photon drag effect not involving material symmetry-breaking[49]. To exclude such an effect, one should measure THz emission with normal incident pump and see whether the THz emission signal is negligible or notable in this geometry. If the THz emission is only from the photon drag effect, one should observe THz emission signal with oblique incident pump and negligible THz emission signal with normal incident pump. We show our THz emission data with oblique incident pump and normal incident pump in Supplementary Note 4. Our THz emission data with normal incident pump is comparable to that with 45 deg oblique incident pump, which cannot be explained by the photon drag effect. Moreover, our pump fluence dependent CPGE effect as presented in Fig. 3a of our main text shows a clear linear to non-linear saturation, which cannot be explained by the photon drag effect either. Therefore, the photon drag effect can be safely ruled out in our case. Additionally, the photo-Dember effect and photo-thermoelectric effect can be ruled out too. They both are pump polarization independent and is directed perpendicular to the sample surface, so that the normal incidence geometry cannot measure it[50,51].

### Computational method and model system

The first-principles density functional theory (DFT) were performed using projector augmented wave (PAW)[52] pseudopotentials with exchange functions Perdew-Burke-Ernzerhof (PBE)[53] implemented in the VASP package[36]. In this work, we use Γ-centered k-point mesh $3 \times 3 \times 2$ and set plane-wave cutoff energy as 400 eV. The spin-orbit coupling (SOC) effect is included in all calculations. $(TaSe_4)_2I$ forms a tetragonal chiral crystal structure with space group $I422$ (No. 97) under ambient conditions, with lattice parameters $a = 9.531 Å$ and $c = 12.824 Å$. In order to obtain crystal structure of non-CDW, we fully relaxed the crystal structure in both the lattice constants and atomic positions until the force on each atom was smaller than 0.01 eV/Å. By distorting along the lowest negative phonon of non-CDW and then do full relaxation, we obtain the crystal structure of CDW phase with space group $F222$(No.22)[10]. Phonons and infrared spectroscopy are calculated by the finite displacement method via Phonopy and Phonopy-Spectroscopy package[54].

## Data availability

The data that support the findings of this work are present in paper and in the Supplementary Information. Source data are provided with this paper. Additional data related to the paper are available from the corresponding authors upon request. Source data are provided with this paper.

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

## Acknowledgements

This work was supported by the U.S. Department of Energy, Office of Basic Energy Science, Division of Materials Sciences and Engineering (Ames National Laboratory is operated for the U.S. Department of Energy by Iowa State University under Contract No. DE-AC02-07CH11358) (topological analysis and DFT calculation). B.C. was supported by the Laboratory Directed Research and Development project, Ames National Laboratory (coherent phonon and photocurrent spectroscopy). W.X. and Y.-F.G. acknowledge the open project of Beijing National Laboratory for Condensed Matter Physics (Grant No. ZBJ2106110017) and the Double First-Class Initiative Fund of ShanghaiTech University (sample synthesis and basic characterization).

## Author contributions

B.C. and J.W. conceived the project, analyzed the spectroscopy data with the input of L.L., B.-Q.S., M.M., and I.-E.P.; B.C., D.C., and L.L. performed the THz emission measurements; T.J. and Y.-X.Y. performed first-principles DFT calculations and topological analysis; W.X. and Y. G. developed samples and performed transport characterizations. The paper is written by J.W., B.C., T.J., and Y.-X.Y. with discussions from all authors. J.W. supervised the project.

## Competing interests

The authors declare no competing interests.
