## [Peer Review File · Nature Communications]

Chirality manipulation of ultrafast phase switches in a correlated CDW-Weyl semimetalREVIEWER COMMENTS

Reviewer #1 (Remarks to the Author):

In this paper, Cheng et al. employed optical excitation to tune and melt various polaronic, axionic, and Weyl semimetallic states or orders within $(\text{TaSe}_4)_2\text{I}$ crystals. $(\text{TaSe}_4)_2\text{I}$ is a complex electronic material exhibiting a coexistence of charge density wave (CDW) and polaronic orders, and it is also chiral. I found this paper to be exceptionally intriguing.

By characterizing the anisotropic response of the circular photogalvanic effect (THz emission due to circularly polarized light), the authors investigated the magneto-electric response arising from the axionic system and its transition to a Weyl semimetal at various excitation fluences.

This paper is remarkably novel and offers several significant contributions to the field, particularly in understanding photoinduced phase transitions, exploring hidden quantum states in complex materials, and characterizing novel orders like the axionic insulator phase in a complex, correlated electronic material. The results are explained with clarity, making a strong case for its publication in Nature Communications.

Suggestion: It would be beneficial to include a brief explanatory paragraph in the supplementary information regarding axionic insulators and their emergence in $(\text{TaSe}_4)_2\text{I}$. This topic represents relatively new physics, and providing some additional details would help readers gain a better understanding.

Minor Question: The paper mentions a polaron gap of 0.2 eV, and it suggests that the first melting step at 0.2 mJ/cm^2 corresponds to the partial melting of polarons. However, it would be helpful if the authors could clarify the rationale behind this assignment. Why does a 0.2 eV polaron gap correspond to a 0.2 mJ/cm^2 fluence for melting?

Is this conversation helpful so far?

Reviewer #2 (Remarks to the Author):

The manuscript concerns the terahertz photocurrent response on the paradigmatic $(\text{TaSe}_4)_2\text{I}$ chain-like material. This compound has been intensively studied since several decades but recent analysis places it at the forefront of strongly correlated phenomena.

$(\text{TaSe}_4)_2\text{I}$ undergoes a phase transition at $T_p=263\text{K}$ initially ascribed as a Peierls type with nesting of some parts of the Fermi surface demonstrating non linear transport properties. This transition is in fact a semiconductor-semiconductor transition in which a liquid of polarons at high temperature (resulting from the strong electron-phonon coupling) condensate below T_c in a long range order charge density wave (CDW).

However the chirality associated to the space group (SG97) $I422$ has been recently acknowledged ranging $(\text{TaSe}_4)_2\text{I}$ in the class of chiral Weyl materials with possible axion electrodynamics behavior.

The main results described in the manuscript is the decoupling of the multi components of the order parameter through a two-steps process as a function of the fluence of the excitation: melting of the polaron state, then a progressive melting of the electronic Weyl—CDW phase before reaching the gapless Weyl phase which was hidden by the correlation gaps. The true chiral nature of the Weyl state is obtained by the helicity-dependent photocurrent.

It is a very nice piece of work very, well written, which is naturally worth to be published in Nature Communications.

It remains however some questions on the nature of the low temperature ground state that will need to be more clarified. Assuming Weyl nodes at the Fermi surface, it is assumed that the CDW results from the strong electron-phonon coupling between these multiple nodes with a different chirality, but in final there exists a single CDW with a unique Q vector with a long -range order. Axion physics has been proposed for this chiral Weyl material; but to ascertain the axionic character

of the CDW in (TaSe₄)₂I it is needed to have the signature of the dynamical axionic quasi particles, that may be controversial.

Reviewer #3 (Remarks to the Author):

This manuscript reports an experimental study of light-induced phase transitions in TaSeI, which was recently proposed to be a Weyl semimetal, in which the Weyl nodes are gapped due to a CDW instability. Here the authors study the dependence of the circular photogalvanic effect (CPGE) on the pump light fluence. The main result, as far as I understood, is the observation of three distinct regimes in this dependence, which the authors interpret

as corresponding to a polaronic insulator, an electronic CDW insulator and a gapless Weyl semimetal.

While the paper does report interesting results, I have some concerns about the interpretation of the data.

1. If I look at Fig. 3a, there is indeed a clear change at $F = 0.2 \text{ mJ/cm}^2$. The rest is much less clear, however.

In particular, there is a kink in the low-temperature data at $F = 0.5 \text{ mJ/cm}^2$, which appears to me just as strong

at the one at 0.8 mJ/cm^2 , but is not interpreted as evidence for a phase change by the authors.

2. I think the evidence for gapless Weyl phase at high fluences is a bit too indirect. The authors' logic, as far as I could

understand, is that a nearly-fluence-independent signal is the evidence for Weyl semimetal phase. Is this really the only possibility? Or is it just suggestive of Weyl semimetal? Also, a strong CPGE in a Weyl semimetal requires a chiral

crystal structure, in which the Weyl nodes of opposite chirality appear at different energies. Is this the case in TaSeI?

If yes, this needs to be explicitly mentioned in the paper. If not, then there is no reason to expect a strong CPGE in the Weyl phase.

In summary, the authors need to clarify their data interpretation and analysis before I can recommend publication.

We express our sincere gratitude to the three reviewers for their careful reading and thoughtful comments and suggestions on our manuscript. We are heartened by their unanimous recognition of the novelties and significance of our experimental observations and the chosen topic. Their insightful feedback has undoubtedly contributed to the refinement of our work.

- Reviewer 1 states: “This paper is remarkably novel and offers several significant contributions to the field, particularly in understanding photoinduced phase transitions, exploring hidden quantum states in complex materials, and characterizing novel orders like the axionic insulator phase...”
- Reviewer 2 states: “...the paradigmatic (TaSe₄)₂I... at the forefront of strongly correlated phenomena... It is a very nice piece of work very, well written...”

Both reviewers have recommended the potential publication of our paper after suggesting some remaining improvements and clarifications.

- Reviewer 1 concludes: “...making a strong case for its publication in Nature Communications.”
- Reviewer 2 concludes: “...naturally worth to be published in Nature Communications.”

We have diligently addressed their feedback in the revised paper. We express our sincere gratitude to the reviewers for their valuable contributions to the refinement of our work. While Reviewer #3 also appreciate our high quality experimental observations but require further clarification for the data interpretation. He/she states: “While the paper does report interesting results, I have some concerns about the interpretation of the data.” We have taken this feedback seriously. Enclosed, you will find our point-to-point responses along with the corresponding changes made in the revised manuscript.

Reviewer #1 (Remarks to the Author):

In this paper, Cheng et al. employed optical excitation to tune and melt various polaronic, axionic, and Weyl semimetallic states or orders within (TaSe₄)₂I crystals. (TaSe₄)₂I is a complex electronic material exhibiting a coexistence of charge density wave (CDW) and polaronic orders, and it is also chiral. I found this paper to be exceptionally intriguing.

By characterizing the anisotropic response of the circular photogalvanic effect (THz emission due to circularly polarized light), the authors investigated the magneto-electric response arising from the axionic system and its transition to a Weyl semimetal at various excitation fluences.

This paper is remarkably novel and offers several significant contributions to the field, particularly in understanding photoinduced phase transitions, exploring hidden quantum states in complex materials, and characterizing novel orders like the axionic insulator phase in a complex, correlated electronic material. The results are explained with clarity, making a strong case for its publication in Nature Communications.

Response #1: We appreciate the positive summary and your support for the publication of our work in Nature Communications.

Suggestion: It would be beneficial to include a brief explanatory paragraph in the supplementary information regarding axionic insulators and their emergence in (TaSe₄)₂I. This topic represents

relatively new physics, and providing some additional details would help readers gain a better understanding.

Response #2: Thanks for the suggestion. We have included a new paragraph in the supplementary to explain why there is an axion insulator phase if Weyl semimetal develops charge-density-wave phase. See supplementary information Note 10 (change #1).

Minor Question: The paper mentions a polaron gap of 0.2 eV, and it suggests that the first melting step at 0.2 mJ/cm² corresponds to the partial melting of polarons. However, it would be helpful if the authors could clarify the rationale behind this assignment. Why does a 0.2 eV polaron gap correspond to a 0.2 mJ/cm² fluence for melting?

Response #3: Thank you for your question. Below, we further discuss the underlying physics.

1. **Polaron gap:** (TaSe₄)₂I is beyond a simple CDW-Weyl material. Above the CDW transition temperature $T_{CDW} \sim 260$ K, it is a polaron insulator. Below T_{CDW} , it is a polaron plus CDW insulator. This picture has been established by early ARPES measurements [PRL 87, 216404 (2001), PRL 110, 236401 (2013), Nat. Phys. 17, 381 (2021)]. We reproduced a plot from a recent ARPES measurement of (TaSe₄)₂I [Nat. Phys. 17, 381 (2021)] in Figure R1. Due to the sample growth process of (TaSe₄)₂I crystals, which may introduce defects and self-doping effects, the Fermi level in these crystals is slightly shifted away from the CDW gap region. As shown in Figure R1, either above or far below T_{CDW} , a consistently observable energy gap near the Fermi energy is present, known as the polaron gap induced by strong electron-phonon coupling. Notably, the CDW gap is situated below the polaron gap, as illustrated in Figure R1. Furthermore, it is worth highlighting that ARPES measurements reveal that **the size of the polaron gap remains essentially consistent ~ 0.2 eV**, both above and significantly below the temperature T_{CDW} .

Figure R1 Regular ARPES data of (TaSe₄)₂I, adapted from Nat. Phys. 17, 381 (2021).

2. **Polaron melting fluence $\sim 0.2 \text{ mJ/cm}^2$:** In our THz emission measurements, distinctive fluence dependence is observed above and below T_{CDW} . Specifically, the Circular Photogalvanic Effect (CPGE) photocurrent exhibits one-step and two-step melting behaviors, respectively, as a function of fluence (refer to Figure R2). Interestingly, the melting fluence for the first step, both above and below T_{CDW} , is **very close to each other**, corresponding to the identical size of the polaron gap above and below T_{CDW} . Consequently, based on the established band structure scenario in Figure R1, it is natural to attribute the first melting process up to $\sim 0.2 \text{ mJ/cm}^2$ to the collapse of the polaron gap in $(\text{TaSe}_4)_2\text{I}$, occurring both above and below T_{CDW} .

Figure R2 Fluence dependence of L-CPGE photocurrent.

3. **Potential correlation:** It is intriguing to correlate the 0.2 eV polaron gap vs the 0.2 mJ/cm^2 melting fluence. Polarons serve as quasiparticles to characterize electrons or holes dressed by ionic vibrations. In our study, we employed an ultrafast laser with a photon energy of approximately 1.5 eV to photoexcite $(\text{TaSe}_4)_2\text{I}$. Essentially, the high energy of a single laser photon is more than sufficient to break one polaron, initiating a process where the resulting hot electrons heat up phonons and further contribute to the polaron-breaking mechanism. Consequently, the fluence required to fully melt the polaron gap, i.e., to break all polarons, becomes contingent on the polaron density, band structure, and electron-phonon coupling of $(\text{TaSe}_4)_2\text{I}$, in addition to the size of the polaron gap. Therefore, the melting fluence reflects the number of photons necessary to completely break all polarons. In this context, the observed 0.2 mJ/cm^2 fluence for melting is reasonable. However, determining the exact relationship involves intricate considerations of the polaron density, band structure, and electron-phonon coupling, making it a challenging problem that falls beyond the scope of our current paper.

Reviewer #2 (Remarks to the Author):

The manuscript concerns the terahertz photocurrent response on the paradigmatic $(\text{TaSe}_4)_2\text{I}$ chain-like material. This compound has been intensively studied since several decades but recent analysis places it at the forefront of strongly correlated phenomena.

$(\text{TaSe}_4)_2\text{I}$ undergoes a phase transition at $T_p=263\text{K}$ initially ascribed as a Peierls type with nesting of

some parts of the Fermi surface demonstrating non linear transport properties. This transition is in fact a semiconductor-semiconductor transition in which a liquid of polarons at high temperature (resulting from the strong electron-phonon coupling) condensate below T_c in a long range order charge density wave (CDW).

However the chirality associated to the space group (SG97) $I422$ has been recently acknowledged ranging $(\text{TaSe}_4)_2\text{I}$ in the class of chiral Weyl materials with possible axion electrodynamics behavior.

The main results described in the manuscript is the decoupling of the multi components of the order parameter through a two-steps process as a function of the fluence of the excitation: melting of the polaron state, then a progressive melting of the electronic Weyl—CDW phase before reaching the gapless Weyl phase which was hidden by the correlation gaps. The true chiral nature of the Weyl state is obtained by the helicity-dependent photocurrent.

It is a very nice piece of work very, well written, which is naturally worth to be published in Nature Communications.

Response #4: Thank you for the insightful summary and your support for the publication of our work in Nature Communications. We are encouraged by your assessment.

It remains however some questions on the nature of the low temperature ground state that will need to be more clarified. Assuming Weyl nodes at the Fermi surface, it is assumed that the CDW results from the strong electron-phonon coupling between these multiple nodes with a different chirality, but in final there exists a single CDW with an unique Q vector with a long -range order. Axion physics has been proposed for this chiral Weyl material; but to ascertain the axionic character of the CDW in $(\text{TaSe}_4)_2\text{I}$ it is needed to have the signature of the dynamical axionic quasi particles, that may be controversial.

Response #5: Thank you for your question. Below, we further clarify the underlying physics.

(1) The nature of CDW in the ground state: There have been some nice AREPS and X-ray diffraction work to determine the Fermi surface topology and the Q vector of CDW. Here we elaborate the points of a nice paper to clarify the nature of CDW in $(\text{TaSe}_4)_2\text{I}$ [Ref 1. Nat. Phys. 17, 381 (2021)]. In the supplementary information of this paper, the band structure calculation of $(\text{TaSe}_4)_2\text{I}$ has identified 24 pairs of Weyl points (WPs) in the bulk within 15 meV of the Fermi energy. The coordinates and multiplicities of these 48 WPs are listed in the Table S1 of Ref 1. According to these coordinates of WPs in momentum space, one can calculate the independent nesting vectors \tilde{q} between WPs with opposite chiral charges. These nesting vectors are listed in the Table S2 of Ref 1. X-ray diffraction experiment can directly measure the direction and magnitude of the CDW modulation vectors. Their data are shown in Fig. S9, S10 of that paper. Considering the possibility of sample domains, the CDW wave vector is determined to be $q = [m\eta\frac{2\pi}{a}, n\eta\frac{2\pi}{a}, o\delta\frac{2\pi}{c}]$, here $\eta = 0.027$, $\delta = 0.012$, and $m + n + o \in 2Z$. By comparing the nesting vectors \tilde{q} between WPs (Table S1) with the CDW wave vector q determined by X-ray diffraction, one can see that all of the nesting vectors \tilde{q} between the WPs with opposite chiral charges match integer multiples of the experimentally-observed CDW modulation basis vectors q . The details of these matches are listed in Table S2 of Ref 1. Therefore, all of the WPs with opposite chiral charges can be nested by the experimentally observed CDW modulation vectors, leading to a gapped CDW phase. We have included a brief discussion in the

supplementary information Note 10, aiming to provide clarification on this issue for readers (change #1).

(2) Regarding the axion physics: In a pure Weyl semimetal without CDW, massless Weyl fermions are just Weyl fermions. They do not have relations to axion insulating phase. As a CDW order emerges in a Weyl semimetal to gap the Weyl nodes, the Weyl fermions near the nodes becomes massive. Most importantly, two new collective modes, i.e. amplitude mode and phase mode, from the CDW order parameter will emerge. In fact, the phase mode θ of CDW order, which is also called CDW phason, is the real dynamical axionic quasiparticle. Weyl fermions themselves are not axionic quasiparticles. The collective phase mode is. Formally, this phase mode θ is termed as axion field or axion mode. It is coupled to the electromagnetic field in the form of $\theta \mathbf{E} \cdot \mathbf{B}$, which has the same form of the axion in the high-energy physics coupling to the electromagnetic field. Indeed, the axion insulator physics in $(\text{TaSe}_4)_2\text{I}$ has been claimed [Nature 575, 315 (2019)] following the general framework of CDW-Weyl semimetal in [PRB 87, 161107(R) (2013)]. Publishing our paper provide motivations to explore the dynamic signatures of axions in $(\text{TaSe}_4)_2\text{I}$ which is still an open question. We appreciate the referee's insightful point and add one sentence in the concluding paragraph "The revealed dynamic strategy of ultrafast and topological phase switching warrants an in-depth investigation to elucidate the dynamic signatures of axion quasiparticles." (Change #2) We also added a new paragraph in the supplementary to explain why there is an axion insulator phase if Weyl semimetal develops charge-density-wave phase (change #1). Please refer to supplementary information Note 10.

Reviewer #3 (Remarks to the Author):

This manuscript reports an experimental study of light-induced phase transitions in TaSeI, which was recently proposed to be a Weyl semimetal, in which the Weyl nodes are gapped due to a CDW instability. Here the authors study the dependence of the circular photogalvanic effect (CPGE) on the pump light fluence. The main result, as far as I understood, is the observation of three distinct regimes in this dependence, which the authors interpret as corresponding to a polaronic insulator, an electronic CDW insulator and a gapless Weyl semimetal. While the paper does report interesting results, I have some concerns about the interpretation of the data.

Response #6: Thank you for the positive feedback from the reviewer, acknowledging the interest in our work.

1. If I look at Fig. 3a, there is indeed a clear change at $F = 0.2 \text{ mJ/cm}^2$. The rest is much less clear, however. In particular, there is a kink in the low-temperature data at $F = 0.5 \text{ mJ/cm}^2$, which appears to me just as strong as the one at 0.8 mJ/cm^2 , but is not interpreted as evidence for a phase change by the authors.

Response #7: Thank you for your question. It is a good point, given that experimental noise is difficult to avoid in pump-fluence dependent measurements. However, we can confidently assert that our interpretation of fluence dependence data is robust for two reasons.

First, to begin with, we focus on global behaviors and robust features guided by physics insights. For the sake of convenience, we have reproduced our main result in Figure R2 below. We have added thick lines to guide the changes of CPGE with pump fluence in Fig. R2a. The CPGE at 5 K exhibits two rise steps: a step with a larger slope below 0.2 mJ/cm^2 and a step with a smaller slope between 0.2 to 0.8

mJ/cm². At 0.2 mJ/cm², the CPGE magnitude is approximately 2.5 a.u., while at 0.8 mJ/cm², the CPGE magnitude is approximately 4 a.u. The CPGE magnitude increases by 60% from 0.2 to 0.8 mJ/cm², which is a notable increase. The two-step rise of CPGE at 5 K is a global feature that is consistent with the existence of the two gaps, i.e., the polaron and CDW gaps. However, the kink at 0.5 mJ/cm² most likely arises from experimental noise, which is difficult to fully avoid. We did not interpret it as evidence of a phase change.

Second, we relied on a controlled analysis to examine the ratio L-CPGE/LPGE. On the one hand, it offers a direct measure of genuine band chirality in the light-induced states, independent of the pump fluence and temperature employed. Longitudinal photogalvanic effect (LPGE) current, unlike CPGE, is not sensitive to the chirality of the Weyl points. On the other hand, the LPGE experiences the same laser fluctuations and other experimental noise as the CPGE. The ratio allows us to improve the signal-to-noise ratio of our measurement. We examine carefully the L-CPGE/LPGE ratio in Fig. R2b which continues its sharp increase above 0.4 mJ/cm² in the low-temperature phase with the CDW gap, and then saturates to a plateau value of approximately 1.5 above 0.8 mJ/cm² (marked by a black dashed line). This dataset shows more clearly that a distinct change indeed occurred around 0.8 mJ/cm². At fluence = 0.5 mJ/cm², there is no anomaly to indicate special phase changes emerge.

Figure R2 Fluence dependence of L-CPGE photocurrent.

2. I think the evidence for gapless Weyl phase at high fluences is a bit too indirect. The authors' logic, as far as I could understand, is that a nearly-fluence-independent signal is the evidence for Weyl semimetal phase. Is this really the only possibility? Or is it just suggestive of Weyl semimetal?

Response #8: Thank you for your question. There are three main reasons for the formation of the gapless Weyl phase at a fluence of approximately 0.8 mJ/cm².

First, we use the temperature and fluence dependence of correlation gaps that are elucidated by our THz emission dataset. In the well-established equilibrium state of (TaSe₄)₂I, discussed in **Response #5**, the compound exhibits a correlated CDW-Weyl phase. This system manifests two distinct phases:

a high-temperature phase, behaving as a polaron insulator with a single gap, and a low-temperature phase, functioning as a polaron plus CDW insulator with two gaps. Our fluence-dependent Circular Photogalvanic Effect (CPGE) data in Fig. R2 uncovers that the correlated gaps in (TaSe4)2I undergo multiple melting stages that are temperature-dependent. The observation of one-step melting from the CPGE current at high temperatures aligns with the melting of the polaron gap. The full melting of the polaron gap results in the recovery of the thermal Weyl phase above $T_{CDW} \sim 260K$ (aqua circles in Figures R2a). In contrast, the observation of two-step melting from the CPGE current at low temperatures corresponds to the melting of not only the polaron but also the electronic CDW gaps (blue rectangles in Figure R2a). The closure of the CDW gap naturally signifies the transition to the gapless Weyl semi-metal phase.

Second, as the electronic CDW gap is completely melted, (TaSe4)2I transitions into a transient Weyl phase characterized by a nearly fluence-independent Circular Photogalvanic Effect (CPGE) current. This is a robust conclusion from that fact that L-CPGE current has a direct correlation to the band chirality of the light-induced Weyl nodes. Our examination of the CPGE/LPGE ratio depicted in Fig. R2b, as discussed in our **response #7**, serves as a direct measure of the genuine band chirality in the light-induced states. Importantly, this ratio remains independent of the pump fluence and temperature employed. Notably, the ratio exhibits a clear saturation above 0.8 mJ/cm^2 (indicated by a red dashed line), which naturally corresponds to the complete closure of the CDW gap and the formation of Weyl semimetal.

Third, the scenario of CDW gap closure aligns seamlessly with the well-established literatures. Notably, previous studies on CDW melting dynamics using ultrafast pump-probe spectroscopy, as exemplified by [PRL 102, 066404 (2009)], have offered crucial insights into the closure of the CDW gap and subsequent recovery of the metallic phase, without the necessity of directly measuring the band structure. These investigations, conducted by analyzing the fluence dependence of pump-probe amplitude and relaxation time, serve as a foundation for our work. Similar to these earlier studies, the melting or closure of the CDW gap in our research is detected through the photoexcited THz CPGE photocurrent. This observation shares a common physics origin with the direct measurement of energy gap closure. Leveraging this well-established knowledge, we present a simple yet effective experimental approach that allows for the accurate access (via an 800 nm ultrafast laser pump), tuning (through pump fluence), and characterization (via THz L-CPGE) to directly measure the chirality of the light-induced phase in this intriguing correlated topological material. Importantly, this approach can be broadly applied to other correlated systems

In summary, our conclusion regarding the scenario of light-induced Weyl nodes is supported by comprehensive evidence, including the CPGE photocurrent across the entire fluence regime and the distinctive temperature dependence of multi-step photocurrent generation and chirality saturation, along with the ultrafast melting dynamics of CDW gaps. Our interpretation stands out as the most natural one, aligning not only with the established scenarios of (TaSe4)2I at equilibrium but also effectively explaining all the observed ultrafast data.

We leave it to the scientific community to assess if there are alternative scenarios capable of explaining all the observations. Our current interpretation upholds scientific rigor based on the spirit of Occam's Razor. We appreciate your reconsideration and support in this matter.

Also, a strong CPGE in a Weyl semimetal requires a chiral crystal structure, in which the Weyl nodes of opposite chirality appear at different energies. Is this the case in TaSeI? If yes, this needs to be

explicitly mentioned in the paper. If not, then there is no reason to expect a strong CPGE in the Weyl phase.

Response #9: The Reviewer's question is well received. $(\text{TaSe}_4)_2\text{I}$ indeed host Weyl nodes at different energies. $(\text{TaSe}_4)_2\text{I}$ is a quasi-1D body-centered tetragonal chiral lattice structure in SG97 (I422) that lacks improper rotation symmetries, such as inversion or mirror symmetry [Nat. Phys. 17, 381 (2021)]. Band structure calculation shows there are 24 pairs of Weyl points near the Fermi energy. All these Weyl points are located within a close vicinity of the $k_z = \pm\pi/c$ planes [Nat. Phys. 17, 381 (2021)]. The CDW phase transition will gap all Weyl points below the Fermi energy. Because the Weyl points with opposite chiral charges appear at different energies, $(\text{TaSe}_4)_2\text{I}$ is predicted to intrinsically carry circular photogalvanic effect (CPGE) [Nat. Mater. 17, 978 (2018), Nat. Phys. 17, 381 (2021)]. This information had been included in the Method section of our manuscript (see Method, L-CPGE in $(\text{TaSe}_4)_2\text{I}$).

In summary, the authors need to clarify their data interpretation and analysis before I can recommend publication.

Response #10: We have addressed the questions raised by the Reviewer and appreciate the Reviewer's reconsideration and support.

List of changes:

1. We added a new section into the supplementary information. See Note 10.
2. We added a new sentence in the concluding paragraph "The revealed dynamic strategy of ultrafast and topological phase switching warrants an in-depth investigation to elucidate the dynamic signatures of axion quasiparticles."
3. We update some references in the original manuscript.
4. We added one sentence in the theoretical discussion "The insights of the mode-selective phononic coupling in strongly correlated materials expands upon recent findings related to light-induced phononic symmetry changes observed in various materials..."

REVIEWERS' COMMENTS

Reviewer #1 (Remarks to the Author):

Dear Editor,

I reviewed the response of the authors to my question and found the answer satisfying. I think this is a very high quality work.

Therefore I recommend its publication in Nature Communications.

Reviewer #3 (Remarks to the Author):

The authors have addressed all my comments in a satisfactory manner. I believe the paper may be published in its present form.

Reviewer #1 (Remarks to the Author):

Dear Editor,

I reviewed the response of the authors to my question and found the answer satisfying. I think this is a very high quality work.
Therefore I recommend its publication in Nature Communications.

Reviewer #3 (Remarks to the Author):

The authors have addressed all my comments in a satisfactory manner. I believe the paper may be published in its present form.

We are happy for referees' agreement on publication.

Additional changes listed below

- 1) We have significantly shortened the abstract to fulfill the format request from Nature Communications.
- 2) We added a short discussion on the debate of the existence of an axion insulator phase in $(\text{TaSe}_4)_2\text{I}$ into Supplementary note 10, following the suggestion from the editor.
- 3) We added two new theoretical references, i.e., ref. 16, 17 into the references of main text to back up the physics of axion insulator phase in CDW-Weyl semimetals.